# Ganglioneuroma in Head and Neck: A Case Report of a Laryngeal Ganglioneuroma and a Systematic Review of the Literature

**DOI:** 10.3390/cancers16203492

**Published:** 2024-10-15

**Authors:** Angela Gasparini, Serena Jiang, Riccardo Mani, Tiziana Tatta, Oreste Gallo

**Affiliations:** 1Department of Otorhinolaryngology, Careggi University Hospital, Largo Brambilla 3, 50134 Florence, Italy; gaspariniangela06@gmail.com (A.G.); oreste.gallo@unifi.it (O.G.); 2Department of Pathology, Careggi University Hospital, Largo Brambilla 3, 50134 Florence, Italy; 3Department of Clinical and Experimental Medicine, University of Florence, 50134 Florence, Italy

**Keywords:** cervical ganglioneuroma, head and neck, neuroblastic tumors, ganglioneuroma management, laryngeal tumors

## Abstract

This research focuses on understanding a rare, benign tumor that derives from the autonomic nervous system, called ganglioneuroma (GN). Our study reports a unique case of a woman with GN originating in her larynx, which caused dysphonia and dysphagia. After an evaluation of the neck imaging, surgical intervention was proposed, and the tumor was successfully removed via a cervical approach. We hope to raise awareness about this unusual tumor location, improve early diagnosis, and guide future treatment.

## 1. Introduction

Ganglioneuroma (GN) is a rare, benign, and encapsulated tumor of nervous tissue originating from the sympathetic ganglion [1]. GN is classified as part of the neuroblastic tumor family, and it is composed of Schwann and ganglion cells [2]. First described by Loretz in 1870 and later by Stout in 1947, this tumor has a higher prevalence in females. It is rarely associated with Von Recklinghausen disease [3].

This neoplasm is commonly localized in the posterior mediastinum, retroperitoneum, and adrenal glands, and it rarely occurs in the head and neck (HN) region, where it typically presents as a painless neck mass [1,2,3]. Due to the indolent and slow-growing nature of these lesions, early detection is challenging, and diagnosis is often delayed until symptoms arise from the compression of surrounding structures. It is typically identified either as a noticeable neck swelling or mass or through incidental findings during radiological investigations performed for unrelated reasons.

Various imaging techniques are useful for cervical masses because they provide information on their size, location, composition, and relation to adjacent structures, but they often lack specificity in distinguishing GN from other neurogenic tumors, so they are mostly used for the follow-up. Fine-needle aspiration biopsy (FNAB) can yield useful diagnostic details, although histopathologic confirmation may not always conclusively identify the condition.

As a result, the definitive diagnosis of GN is generally established only after surgical resection, which allows for thorough histopathological examination. Surgery represents the first therapeutic choice, but complications can occur due to the proximity of these tumors to vascular and nervous structures in the cervical region [4].

Herein, we describe a case of GN in an adult patient with worsening dysphonia and dyspnea that manifested as a swollen mass in the left lateral neck region. She underwent surgical resection of the mass, and subsequent histopathological analysis confirmed the diagnosis of larynx-originating GN. In light of this case, we also provide a comprehensive review of the existing literature, highlighting current approaches to the diagnosis, management, and prognosis of these tumors in the HN region.

## 2. Case Report

A 43-year-old Caucasian female, with a smoking history of 20 pack-years and no comorbidities, reported worsening dysphonia associated with dyspnea for approximately one year.

Fibrolaryngoscopy (FLS) evaluation revealed a voluminous right-sided mass at the hypopharyngeal level, obstructing visualization of the glottic plane and covered by intact mucosa. No cervical lymphadenopathies were clinically evident (Figure 1).

Although Magnetic Resonance Imaging (MRI) of the neck was recommended, the patient underwent computed tomography (CT) with and without contrast because of claustrophobia. CT revealed a voluminous, round, heterogeneous solid mass measuring 44 × 33 mm in the axial planes, extending cranially by 65 mm, localized in the paramedian right-sided hypopharynx, and involving the posterolateral parapharyngeal space (Figure 2). It abutted the right epiglottis without a cleavage plan and extended inferiorly, displacing the glottic plane and aryepiglottic fold and infiltrating the cricoid cartilage and the right thyroid oblique lamina. The upper right pole of the thyroid gland was not cleavable. Enlarged ipsilateral lymph nodes (LNs) were noted.

Diagnostic microlaryngoscopy with multiple biopsies of the lesion and tracheotomy was performed under general anesthesia. Histological examination showed a paracellular neoplasm composed of spindle elements without atypia, with positive immunohistochemistry for S100 and SOX10. The findings were suggestive of a nerve sheath neoplasm without histological features of aggressiveness.

Therefore, we performed a surgical excision through a right cervical access. We removed the upper part of the right thyroid cartilage, including its superior horn, and the homolateral pharyngeal constrictor muscles were removed; this allowed the complete removal of the encapsulated, oval (50 × 30 × 30 mm), yellowish-white mass, with its origin at the cricothyroid notch and the epiglottis. The glottis and both arytenoids were preserved.

At the microscopic examination (Figure 3), the predominant portion of the neoplasm consisted of spindle cells with sporadic cytological atypia, while a minor portion was represented by epithelioid cellularity with a neuroendocrine immunophenotype. Positive staining was observed with S-100, CD56, chromogranin, and synaptophysin. Some elements exhibited abundant polygonal cytoplasm with evident nucleoli and cytoplasmic extensions (ganglion cells). Mitosis, lymphatic/vascular invasion, and necrosis were not documented. Ki-67 was expressed in 5–8% of the neoplastic population. A pathological diagnosis of GN was established.

The tracheal cannula and a nasogastric tube were maintained until the 7th post-operative day, and she was discharged on the 10th post-operative day. The FLS examination showed mild arytenoid edema and reduced mobility of the right arytenoid with preserved airway space and an absence of pathological stasis in the hypopharynx. She underwent speech therapy rehabilitation for the first 3 months post-surgery.

At the 6-month follow-up evaluation, the patient reported a major improvement in dysphonia and an absence of dysphagia. The FLS showed no recurrence and persisting hypomobility of the right vocal cord with good compensation by the contralateral one.

## 3. Material and Methods

### 3.1. Searching Strategy and Selection Criteria

This review was performed in accordance with the PRISMA (Preferred Reporting Items for Systematic Reviews and Meta-Analyses) guidelines and has not been registered. We conducted a literature search using the PubMed and Embase databases in order to identify the relevant studies.

The following keywords were used:Ganglioneuroma AND head and neck (160 articles in English, Italian, or Chinese);Ganglioneuroma AND larynx (7 articles in English, Italian, or Chinese);

Only studies describing the localization, symptoms, and therapeutic management of patients with HN GN were included; articles were excluded based on the following criteria: localization outside the HN region, papers with incomplete data and reviews, and those that were not written in English, Italian, or Chinese.

### 3.2. Data Collection

The title and abstract of the selected papers were carefully read according to the inclusion and exclusion criteria, and duplicates were removed. Two reviewers (S.J., A.G.) independently extracted data from each study, which were reviewed for consistency among the authors, and any discrepancies were resolved by consensus. The full text of the included studies was then read in order to extract the following data:Reference: first author, year of publication;Age and sex;Localization and size;Symptoms;Therapeutic iter and surgical complications;Follow-up duration.

### 3.3. Quality Assessment and Statistical Methods

The quality and the risk of bias of the articles included in the systematic review were evaluated by the Quality In Prognosis Studies (QUIPS) tool, with any discrepancies resolved by consensus by the first two authors (S.J., A.G.). Visualization of the risk-of-bias assessments was performed by creating a traffic lights plot and a weighted bar plot using the Robvis tool.

## 4. Results

We systematically reviewed cases of HN GN reported in the literature up to January 2024, following the PRISMA guidelines.

A total of 160 records for HN GN were identified from a primary literature search. After the removal of duplicates and by applying the aforementioned criteria, a total of 66 publications were selected. Papers were then screened by reading the titles and abstracts, and 58 manuscripts were deemed eligible for possible inclusion. After reading the full texts, 4 articles were excluded because of insufficient or incomplete data, and 1 review and 3 articles were excluded because of the GN localization; only 58 studies eventually met the inclusion criteria. The flowchart presenting our literature search strategy is shown in Figure 4. The included studies are summarized with their main characteristics in Table 1.

The majority of the population included in our study (N = 65) was under 30 (67.69%). The average age at presentation was 24 years old, with a median of 19 years, with the maximum reported age in the literature being 71 years. There were no differences between males and females (M = 31, F = 33, NA = 0). In the cervical region, the left side was more frequently involved; the localizations described in the literature were laterocervical, oropharyngeal, and the retropharyngeal space [11,16,30,33,40]. HN GN is usually a single, monolateral lesion, but a case of bilateral neck GN was also reported [44]. None described a laryngeal localization. Patients presented predominantly with cervical swelling, rarely with dysphagia or pain.

The postoperative complications were mostly neurological: the most frequently described one was Horner’s syndrome (33.85%), particularly if the chosen surgical approach was transcervical. Less frequently, paralysis of mixed cranial nerves (IX, X, XI, XII) occurred, resulting in symptoms such as dysphonia, dysphagia, and shoulder motor difficulty [2,12,21,22,29,30,34,35,36]. In one case, surgical intervention led to the onset of first-bite syndrome [18].

Their overall risk of bias was judged to be low to moderate, and the traffic lights plot and the weighted bar plot for each domain considered in the QUIPS tool are given in Figure 5 and Figure 6, respectively.

## 5. Discussion

GN is a rare, benign tumor originating from the sympathetic ganglion, composed of Schwann and ganglion cells. It is more common in females and is only rarely linked to Von Recklinghausen disease [1].

GNs are slow-growing, well-differentiated tumors of the autonomic nervous system [24]. They are usually located in the thoracic cavity (60–80%, mainly in the posterior mediastinum), abdominal cavity (10–15%, adrenal gland, retroperitoneum, pelvic, sacral, and coccygeal sympathetic ganglia, and the organ of Zuckerkandl), and less commonly in the cervical region (5%). Other rare locations include the middle ear, parapharyngeal space, skin, orbital space, and gastrointestinal tract [25,26]. To the best of our knowledge, there are not any described cases of laryngeal GNs.

Although Stout (1947) recorded a predominance of this tumor in females [3], the current literature data show no incidence prevalence of the condition based on gender, with a male-to-female ratio of 1:1.

Our results indicate that this condition typically affects young adults, with 67.69% of the patients described as under 30 years old. The average age at presentation was 24 years, with a median age of 19 years, and the oldest age reported in the literature was 71 years.

Typically, GNs grow slowly and cause symptoms only when they reach a considerable size or compress nearby structures. In cases where they exhibit low neuroendocrine activity, they may lead to symptoms such as diarrhea, hypertension, virilization, and myasthenia gravis. Tumors located in the HN region usually present with neck swelling, hoarseness, and dysphagia; large GNs can compress peripheral nerves in the cervical area, resulting in Horner’s syndrome [23].

This symptomatology may be common to other expansive head–neck pathologies that cause compression of the lateral–cervical structures. The differential diagnosis, in fact, includes paragangliomas, lymphomas, carcinomas, thyroid masses, neurofibromas, and salivary gland tumors. Therefore, a thorough medical history examination is essential to rule out systemic symptoms that could indicate lymphatic neoplasms or syndromic conditions (such as neurofibromatosis). Additionally, laboratory testing to exclude abnormalities in thyroid hormone levels is indispensable [18].

The largest GN reported in the literature measured 160 × 60 × 190 mm, and the patient reported neck swelling and sleep apnea [15]. The symptoms and the localization of our patient were uncommon: she reported worsening dysphonia and dyspnea due to its laryngeal origin.

Advanced imaging techniques play a crucial role in assessing the extent and characteristics of GN, providing information on mass size, location, composition, and its relation to adjacent structures. A combination of imaging modalities, including ultrasound (US), CT, MRI, positron emission tomography-computed tomography (PET-CT), and fine-needle aspiration biopsy (FNAB), helps in accurate diagnosis and treatment planning [20]. US provides an initial evaluation and can identify the location and size of the tumor, while FNAB, although not always definitive, may provide valuable diagnostic information in some cases. CT scans offer detailed visualization of the tumor’s location, size, and morphology, typically presenting as well-defined, low- or medium-density masses with calcifications. MRI is particularly useful for soft tissue characterization, delineating the tumor’s relationship with surrounding structures and depicting features such as the “vortex sign” on T2-weighted images. PET-CT may offer insights into the metabolic activity of the tumor. However, it is essential to note that characteristic features of GNs can overlap with other neurogenic tumors, requiring a comprehensive approach that combines clinical findings, imaging studies, and histopathological evaluation for accurate diagnosis and treatment decision-making [58].

Making an accurate preoperative histological diagnosis of GNs poses challenges since FNAB is not consistently definitive. Histologically, GNs are composed of ganglion cells, Schwann cells, and fibrous tissues. Their differential diagnosis includes neuroblastoma and ganglioneuroblastoma based on neuroblastic differentiation and Schwannian stroma development [17].

The treatment strategy for GN remains controversial. Surgical resection is not mandatory because the prognosis is generally favorable and post-operative complications may occur, mostly neurological, albeit being resolved within months. It should be considered in cases that occur with neurological symptoms, hormonal imbalances, or significant physical deformities. And in some cases when a neurogenic tumor with malignant potential is suspected, surgery represents the only way to obtain a definitive diagnosis.

Surgical resection is the mainstay of treatment for parapharyngeal space tumors, with various surgical approaches developed to balance maximal tumor removal with minimal damage to surrounding structures [4]. The lateral cervical approach is commonly employed due to its wide surgical field and better exposure to important neck structures. In our review, three cases preferred the transoral approach, while in two cases, a combined approach was used.

When deciding between surgical intervention and a “wait and see” approach, it is important to consider the most common postoperative complications. Postoperative complications are primarily neurological, with Horner’s syndrome being the most commonly reported (33.85%), especially when a transcervical surgical approach is used. Less commonly, paralysis of mixed cranial nerves (IX, X, XI, XII) has been observed, leading to symptoms such as dysphonia, dysphagia, and shoulder motor difficulties [2,12,21,22,29,30,34,35,36]. In one instance, surgery resulted in the development of first-bite syndrome [18].

Recurrence, although rare, can happen, emphasizing the importance of thorough surgical management and long-term follow-up [35].

## 6. Conclusions

GN is a benign tumor that can occur in various parts of the body. The HN region is an uncommon site and, to the best of our knowledge, the laryngeal location has not been previously reported. Due to its indolent nature, GN often presents as an incidental finding or as a symptomatic mass due to its size or compression of surrounding structures. Imaging and FNAB help us with the diagnosis. Surgery should only be considered in cases with neurological symptoms, hormonal imbalances, or significant physical deformities because it is associated with potential postoperative neurological complications. Although recurrence is rare, it underscores the importance of vigilant follow-up.

Given the rarity of GN in the HN region, particularly in the laryngeal region, we conducted a systematic search of the literature to establish treatment strategies, but large-scale studies are necessary to assess evidence-based protocols.

Written consent was obtained from the patient for the publication of this study. Our research was conducted in full accordance with ethical principles, including the World Medical Association Declaration of Helsinki (version 2002).

## Figures and Tables

**Figure 1 cancers-16-03492-f001:**
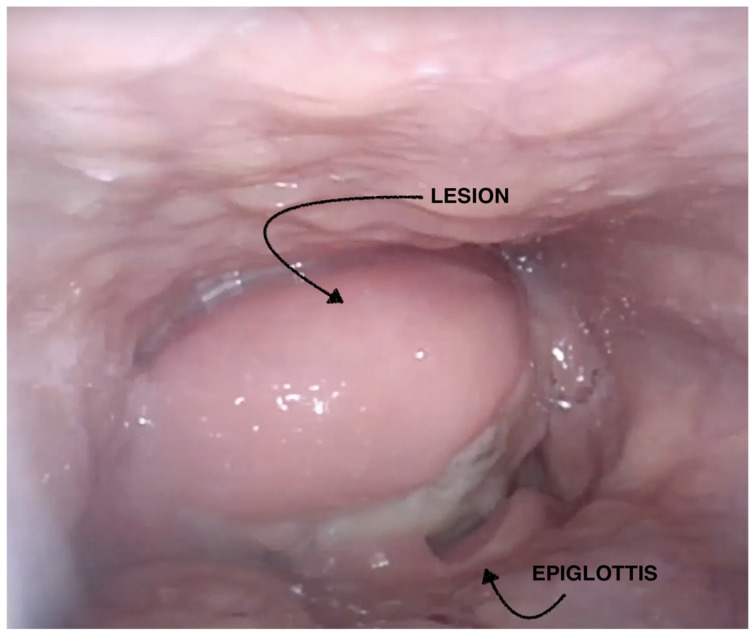
Preoperative fibrolaryngoscopy.

**Figure 2 cancers-16-03492-f002:**
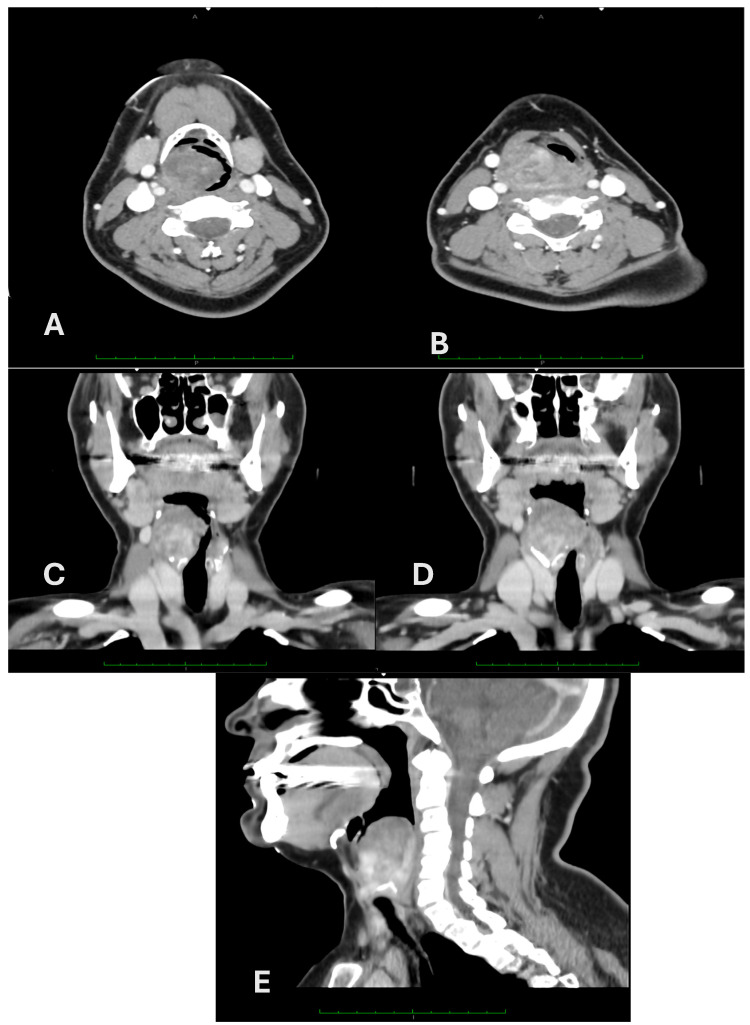
Axial (**A**,**B**), coronal (**C**,**D**), and sagittal (**E**) contrast-enhanced CT scan of the neck.

**Figure 3 cancers-16-03492-f003:**
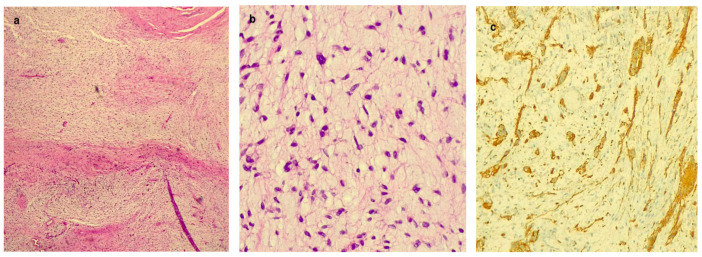
Histological findings of laryngeal ganglioneuroma. (**a**) Hematoxylin and eosin ×25; (**b**) hematoxylin and eosin ×200; (**c**) synaptophysin stain.

**Figure 4 cancers-16-03492-f004:**
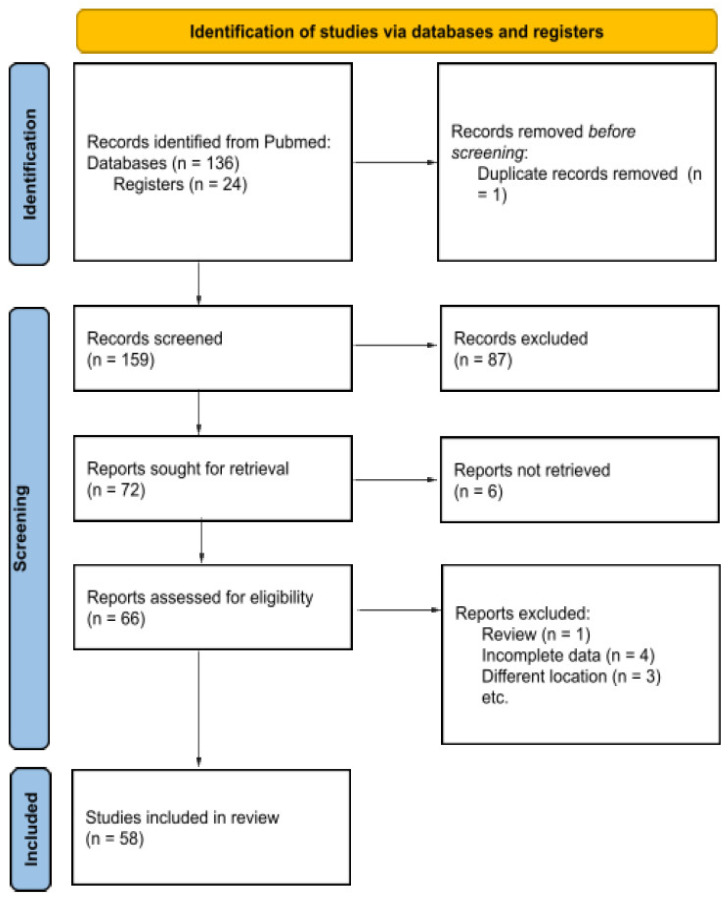
PRISMA 2020 flow diagram for new systematic reviews.

**Figure 5 cancers-16-03492-f005:**
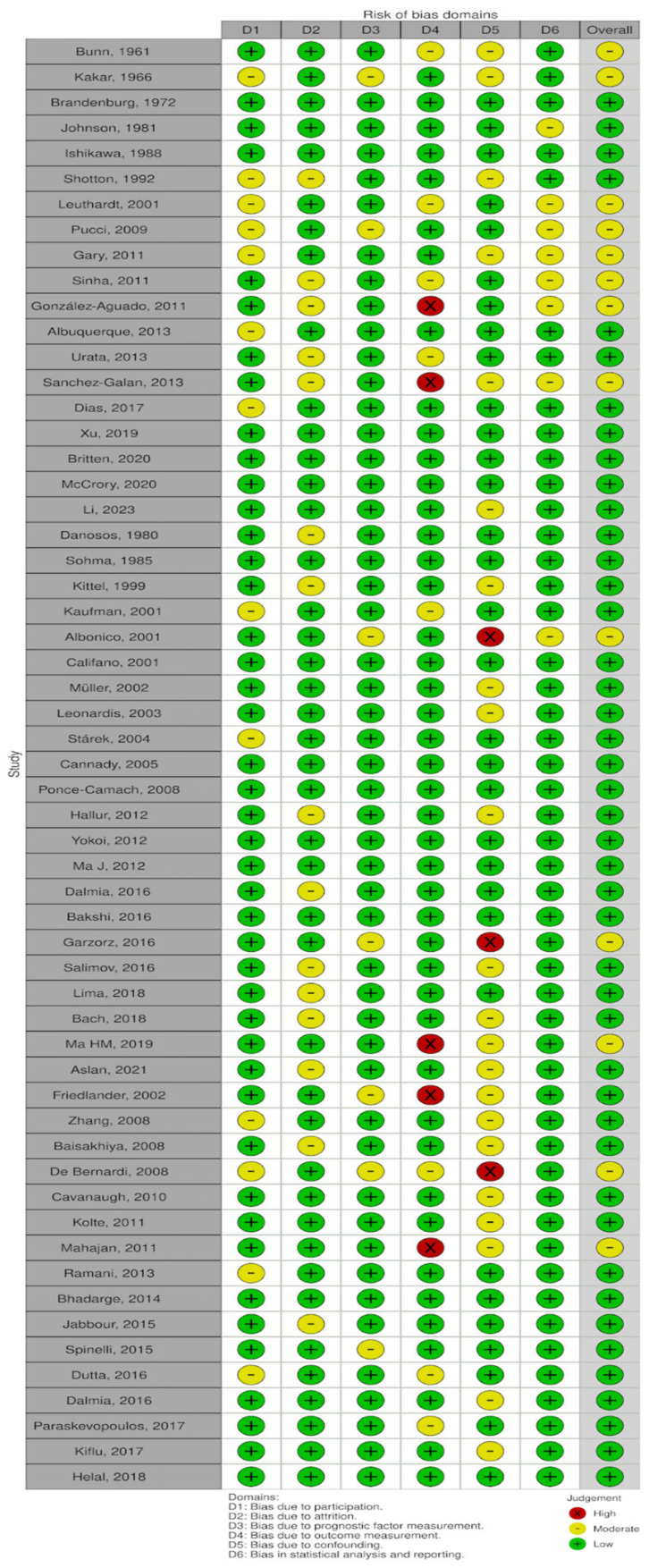
Traffic lights plot [2,3,4,5,6,7,8,9,10,11,12,13,14,15,16,17,18,19,20,21,22,23,24,25,26,27,28,29,30,31,32,33,34,35,36,37,38,39,40,41,42,43,44,45,46,47,48,49,50,51,52,53,54,55,56,57].

**Figure 6 cancers-16-03492-f006:**
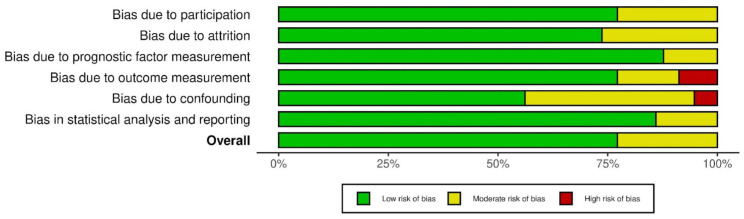
Weighted bar plot.

**Table 1 cancers-16-03492-t001:** Clinicopathological features of cervical ganglioneuromas described in the literature. NA Not Available.

n°	Authors	Age	Gender (M = Male, F = Female)	Size (mm)	Side, Location	Symptoms	Surgery (0 = None; 1 = Transcervical; 2 = Transoral; 3 = Combined)	Complications	Follow-Up (Months)
1	Bunn, 1961 [5]	51	M	53 × 32 × 30	right, neck	pain, swelling, dysphagia	1	Horner’s syndrome	NA
2	Kakar, 1966 [3]	9	M	27 × 10	left, neck	swelling	1	none	NA
3	Brandenburg, 1972 [6]	21	M	60 × 80	right, neck	swelling	1	Horner’s syndrome	NA
4	Johnson, 1981 [7]	65	F	20 × 20	left, neck	swelling	1	hoarseness	NA
5	Ishikawa, 1988 [2]	9	M	NA	left, neck	swelling	1	recurrent nerve palsy	NA
6	Shotton, 1992 [8]	6	M	40 × 30	left, neck	swelling	1	Horner’s syndrome	NA
7	Leuthardt, 2001 [9]	5	F	NA	left, neck	stridor	1	cortical blindness	NA
8	Pucci, 2009 [10]	25	F	22 × 22 × 47	right, neck	swelling	1	Horner’s syndrome	NA
9	Gary, 2011 [11]	42	F	55 × 32 × 10	right, retropharyngeal	neck stiffness, dysphagia, dysphonia	1	Horner’s syndrome	NA
10	Sinha, 2011 [12]	22	M	90 × 70 × 40	left, neck	swelling	1	hypertension, hoarseness, nasal regurgitation	36
11	González-Aguado, 2011 [13]	41	F	NA	right, neck	na	1	NA	11
12	Albuquerque, 2013 [14]	5	F	64 × 45 × 28	left, neck	swelling	1	Horner’s syndrome	NA
13	Urata, 2013 [15]	18	M	160 × 60 × 190	right, neck	swelling, sleep apnea	1	none	7
14	Sanchez-Galan, 2013 [4]	36	M	NA	NA	stridor, hoarseness	1	none	NA
15	Dias, 2017 [16]	71	M	27 × 10 × 27	left, retropharyngeal	none	0	none	13
16	Xu, 2019 [17]	12	M	46 × 17 × 16	left, neck	swelling	1	Horner’s syndrome	8
17	Britten, 2020 [18]	30	F	50 × 40	right, neck	swelling	1	first bite syndrome	1.5
18	McCrory, 2020 [19]	49	F	13 × 8 × 3	left, neck	none	1	Horner’s syndrome	8
18	Li, 2023 [20]	58	M	88 × 48 × 48	left, neck	pharyngeal foreign body sensation	1	NA	12
20		22	M	60 × 40 × 35	right, neck	swelling	3	NA	114
21	Danosos, 1980 [21]	42	F	100 × 60 × 30	left, neck	sore throat	1	Horner’s syndrome, vocal cord paralysis	4
22	Sohma, 1985 [22]	57	M	30 × 25 × 20	right, neck	pharyngeal foreign body sensation, hoarseness	1	IX X XII palsy	NA
23	Kittel, 1999 [23]	19	NA	100 × 60 × 40	right, neck	swelling, dysphagia	NA	myosis	NA
24	Kaufman, 2001 [24]	11	F	20	right, neck	swelling	1	Horner’s syndrome	6
25	Albonico, 2001 [25]	27	F	40 × 30	right, neck	swelling, pain	NA	na	NA
26	Califano, 2001 [26]	26	F	60 × 45 × 25	left, neck	sleep apnea	1	Horner’s syndrome, dysphagia	24
27	Müller, 2002 [27]	35	F	35 × 30 × 15	left, neck	pain	2	Horner’s syndrome	NA
28	Leonardis, 2003 [28]	50	M	100 × 70 × 40	left, neck	swelling	1	ptosis	NA
29	Stárek, 2004 [29]	2	F	50 × 35 × 25	right, neck	swelling	1	vocal cord paralysis	24
30	Cannady, 2005 [30]	6	M	60 × 40 × 28	right, neck	swelling	1	Horner’s syndrome	24
31		7	M	60 × 30 × 20	right, neck	oral swelling	2	Horner’s syndrome	156
32		9	F	60 × 50	right, oropharynx	dysphagia	1	IX XII paralyses	108
33	Ponce-Camach, 2008 [31]	5	F	40 × 30	right, neck	swelling	1	Horner’s syndrome	NA
34	Hallur, 2012 [32]	4	F	100 × 64 × 57	right, neck	swelling	1	myosis	18
35	Yokoi, 2012 [33]	19	F	72 × 33 × 11	oropharynx	swelling	2	Horner’s syndrome	60
36	Ma, J., 2012 [34]	4	F	54 × 37 × 33	right, neck	swelling, dysphagia	1	XII paralysis	1
37	Dalmia, 2016 [35]	25	M	50 × 30	left, neck	swelling, hoarseness	1	X paralysis	NA
38	Bakshi, 2016 [36]	10	F	40 × 50	left, neck	swelling	1	XII palsy	12
39	Garzorz, 2016 [37]	3	M	60 × 28 × 17	neck	swelling	1	NA	36
40	Salimov, 2016 [38]	13	F	115 × 55 × 40	left, neck	swelling	1	Horner’s syndrome	12
41	Lima, 2018 [39]	13	F	60 × 35	right, neck	swelling	1	Horner’s syndrome	24
42	Bach, 2018 [40]	17	M	86	right oropharynx	dysphagia	3	NA	NA
43	Ma, H.M., 2019 [41]	35	F	NA	right, neck	headache	2	NA	18
44	Aslan, 2021 [42]	41	M	100 × 50 × 40	left, neck	swelling	2	myosis	12
45	Friedlander, 2002 [43]	28	M	40 × 20 × 43	left, neck	NA	NA	NA	NA
46	Zhang, 2008 [44]	6	M	40 × 30	left, neck	swelling	1	none	NA
47		62	F	80 × 40	left, neck	swelling	1	none	NA
48		57	F	80 × 70	right, neck	swelling	1	none	NA
49		9	M	40 × 20	bilateral, neck	swelling	1	none	NA
50		53	F	40 × 40	right, neck	swelling	1	none	NA
51	Baisakhiya, 2008 [45]	22	M	30 × 25 × 30	left, neck	NA	NA	Horner’s syndrome	NA
52	De Bernardi, 2008 [46]	2	M	NA	neck	NA	NA	Horner’s syndrome	130
53	Cavanaugh, 2010 [47]	41	M	NA	left, neck	none	1	Horner’s syndrome	NA
54	Kolte, 2011 [48]	8	F	50 × 40 × 30	left, neck	swelling	1	NA	NA
55	Mahajan, 2011 [49]	7	M	70 × 55 × 50	left, neck	swelling	1	NA	NA
56	Ramani, 2013 [50]	5	F	50 × 45 × 30	left, neck	swelling	1	none	NA
57	Bhadarge, 2014 [51]	11	F	100 × 55 × 40	left, neck	swelling	1	none	NA
58	Jabbour, 2015 [52]	53	M	32 × 25 × 22	left, neck	swelling	1	none	10
59	Spinelli, 2015 [53]	26	F	NA	neck	swelling	1	none	96
60		37	F	NA	neck	none	1	none	84
61	Dutta, 2016 [54]	1	M	30 × 20	left, neck	swelling	1	NA	NA
62	Dalmia, 2016 [35]	25	M	50 × 30	left, neck	swelling, hoarseness	1	vocal cord palsy	NA
63	Paraskevopoulos, 2017 [55]	17	F	40 × 25 × 10	left, neck	swelling	1	none	NA
64	Kiflu, 2017 [56]	7	F	50 × 70 × 30	left, neck	swelling	1	Horner’s syndrome	NA
65	Helal, 2018 [57]	12	M	46 × 17 × 16	left, neck	swelling	1	none	12

## Data Availability

Data are available in the Pubmed and Embase databases.

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
