# Peer review of "Ganglioneuroma in Head and Neck: A Case Report of a Laryngeal Ganglioneuroma and a Systematic Review of the Literature"

_cancers, 2024, doi:10.3390/cancers16203492_

Round 1

Reviewer 1 Report

Comments and Suggestions for Authors

In this paper Gasparini et al have described a case of laryngeal ganglioneuroma, and have included an accurate review of the literature extended to head and neck region. The case is interesting and rightly reported, and deserves to be published on Cancer. However, due to the rarity of the lesion, the Authors should complete their report by at least one representative histologic image of the tumor, obtained from the surgical specimen.

Author Response

Dear reviewer, thank you for your kind suggestion. I've updated the manuscript with 3 images of the histopathological findings.

Reviewer 2 Report

Comments and Suggestions for Authors

The manuscript "Ganglioneuroma in Head and Neck: A Case Report of a Laryngeal Ganglioneuroma and a Systematic Review of the Literature" is scientifically and clinically relevant and has novelty. However, the authors has 2 different types of manuscript in this version. A case report and a systematic review and this types are completly different. 

- The case report is incomplete and the structure is not respected. A case report has results, discussion and conclusion.

- The systematic review has also some aspects to improve, mainly the discussion. However, PRISMA is a methods and figure 5 is also a method.

The discussion is a combination of the case report and the systematic review but not disccuss the major results of the manuscript. As a suggestion, the authors have 2 different manuscripts. A case report and a systematic review, that are both acceptable for publication. However not in combination.

Comments on the Quality of English Language

Minor editing of English language required.

Author Response

Dear reviewer, thank you for your suggestions. Our article is intended as a systematic review enriched by a rare case of laryngeal GN that presented to our institution. That are many systematic reviews that include a case report right after the introduction part.

Round 2

Reviewer 1 Report

Comments and Suggestions for Authors

The Authors made up the suggested changes. The article is now suitable for publication.